# A meta-analysis of the prevalence, genotype distribution and risk factors for human papillomavirus infection in Nepal

**Prayash Paudel**⊙*, **Asutosh Sah**

Bachelor of Medicine and Bachelor of Surgery, Maharajgunj Medical Campus, Tribhuvan University Teaching Hospital, Institute of Medicine, Nepal, Bagmati Province, Nepal

⊙ These authors contributed equally to this work.
* prayash.784401@mmc.tu.edu.np

## Abstract

This systematic review and meta-analysis estimated the prevalence, genotype distribution, and risk factors for human papillomavirus infection among Nepalese women. A total of 8 cross-sectional studies with 6,082 participants were included in this review. For this review, the eligibility criteria included studies reporting HPV prevalence or genotype-specific rates among Nepalese women; thus, database sources were searched up to December 2024. The quality of the studies was assessed, and pooled estimates were computed via random effects models. The pooled prevalence of HPV infection among Nepalese women was 8.31%, with a genotype-specific prevalence of 2.62% for HPV-16 and 1.25% for HPV-18. HPV infection was significantly associated with those whose husbands had multiple sexual exposures. Other factors, including smoking status, educational status, and the use of contraceptives, were not significantly associated. Most analyses showed moderate to high heterogeneity. This review highlights the moderate burden of HPV infection in Nepal; hence, targeted public health intervention through vaccination and screening programs is recommended. Limitations include heterogeneity of studies and an inability to draw causal interference because all the data were cross-sectional. The findings highlight the need for region-specific strategies to combat HPV-related diseases in Nepal and underscore the importance of future research to address existing knowledge gaps. This review is registered in PROSPERO with registration number CRD42025632655.

## Introduction

Sexually transmitted infection caused by human papillomavirus (HPV) is common worldwide, with high-risk types—particularly HPV-16 and HPV-18—strongly linked to cervical cancer and other anogenital malignancies. In 2020, cervical cancer was the fourth leading cancer among women worldwide, accounting for approximately 604,000 new cases and 342,000 deaths [1].

**Data availability statement:** All relevant data are within the manuscript and its Supporting Information files.

**Funding:** The author(s) received no specific funding for this work.

**Competing interests:** The authors have declared that no competing interests exist.

Cervical cancer is a major public health problem in Nepal. A population-based study from rural Nepal reported an overall prevalence of HPV of 14.4%, with 7.9% attributed to high-risk types [2]. A study among urban and semiurban women in the south-central plains of Nepal reported a prevalence of high-risk HPV at 8.6% [3]. These studies revealed the high burden of infection due to HPV across various regions of the country.

Despite the availability of preventative measures against HPV and cervical screening, coverage in Nepal remains inadequate; a 2022 study across five tertiary hospitals in Kathmandu reported that only 1.5% of women had received the HPV vaccine and just 22.2% had ever undergone a Pap smear test for cervical cancer screening. Several studies have identified sociocultural barriers, including misconceptions regarding screening and prevalent stigma, which act as barriers to the participation of women in screening programs [4]. Lack of education about cervical cancer and its prevention adds to the dismal state of uptake for screening services [5]. Geographical challenges and financial constraints also hinder access to healthcare facilities providing such services [4]. Addressing multifaceted barriers will surely help improve the coverage of HPV vaccination and screening for cervical cancer in Nepal. A study carried out in midwestern Nepal revealed a higher prevalence of high-risk HPV infection (11.7%) than in other regions; HPV-16 was the most prevalent genotype [6]. Therefore, region-specific strategies for addressing diseases related to HPV are needed.

Understanding the prevalence and genotype distribution of HPV infection in Nepal is important for the development of effective public health strategies, targeted vaccinations, and screening programs. Given the high burden of HPV-related diseases and the low use of preventive measures, a comprehensive review of existing studies on HPV prevalence in Nepal is warranted.

The goal of this systematic review is to comprehensively analyse the prevalence and distribution of HPV infection among Nepalese women and explore the rates of prevalence as presented and compared with global and regional rates. It also aims to identify the major HPV genotypes reported within the country and their prevalence.

This paper therefore systematically reviewed the literature to provide a comprehensive overview of HPV prevalence in Nepal, identify gaps in current knowledge, and generate recommendations for future research and public health interventions, thus contributing to a better understanding of HPV epidemiology in Nepal and to the elaboration of future public health guidelines.

## Methods

### Study protocol

The study protocol, with well-defined methodology and inclusion criteria, was registered on PROSPERO with registration number CRD42025632655.

### Search strategy

This systematic review and meta-analysis followed the principles of the Preferred Reporting Items for Systematic Reviews and Meta-Analyses, as described in the

S4 Appendix and S5 Appendix. The PRISMA diagram detailing the selection process is shown in Fig 1. The PubMed, Embase and Google Scholar databases were searched up to 30th December 2024. Studies were also obtained from supplementary sources, manual searches and other repositories. Cross-references from the published articles were manually searched to retrieve additional literature.

To create an extensive search strategy that encompassed all fields in the records as well as Medical Subject Headings (MeSH words) for broadening the search in an advanced PubMed search, the predefined phrases were identified. For the PubMed database, MeSH terms and relevant keywords were systematically combined via Boolean operators (AND, OR) to identify records. Details of the preliminary search strategy are provided in the S1 Appendix. Similarly, for Embase, Emtree terms and keywords were combined with Boolean operators to ensure the comprehensive retrieval of relevant studies. The search terms included combinations of 'human papillomavirus' and 'Nepal.' In Google Scholar, a similar approach was employed using the keywords 'human papillomavirus' and 'Nepal' to capture additional literature.

Inclusion criteria:

1. Studies involving populations from Nepal.

2. Community-based, hospital-based, clinical, or health facility-based settings where HPV testing was performed.

3. Cohort, case–control, or cross-sectional studies.

4. Studies focused on risk factors such as smoking status, educational status, sexual behaviour, STI history, contraceptive use, and husbands' migration status.

Exclusion criteria:

1. Nonhuman studies (e.g., animal studies, cell line studies).

2. Studies not focused on HPV infection or HPV-related conditions (e.g., studies on other types of cancers or non-HPV-related conditions).

3. Studies focusing on non-Nepali populations.

4. Studies without sufficient data on risk factors associated with HPV.

5. Studies with incomplete or unclear methodologies regarding population selection, exposure variables, or outcomes related to HPV infection.

## Selection of studies

The literature search was performed by PP. The included studies were exported to Google Sheets in compatible format. Duplicate articles were screened manually. Duplicates were then recorded and removed. After removing duplicates, two independent authors, PP and AS, screened the title and abstract of every remaining article. Full-text articles were obtained for the relevant studies satisfying the inclusion criteria. The data were extracted by the two authors, PP and AS, independently. Any disagreements were resolved through discussion and mutual consensus.

## Data extraction

The following data were extracted from the studies: name of the author, year of publication, study design, sample size, mean age of the population, mean age at first marriage, total number of cases, number of cases identified with single infections, multiple infections, and the total number of infections attributed to genotype 16 and genotype 18, smoking status of HPV-infected participants (categorized as smokers and nonsmokers), educational status of HPV-infected participants (literate-those with at least primary education and illiterate-those with no formal education or inability to read and write),

PRISMA 2020 flow diagram for new systematic reviews which included searches of databases and registers only

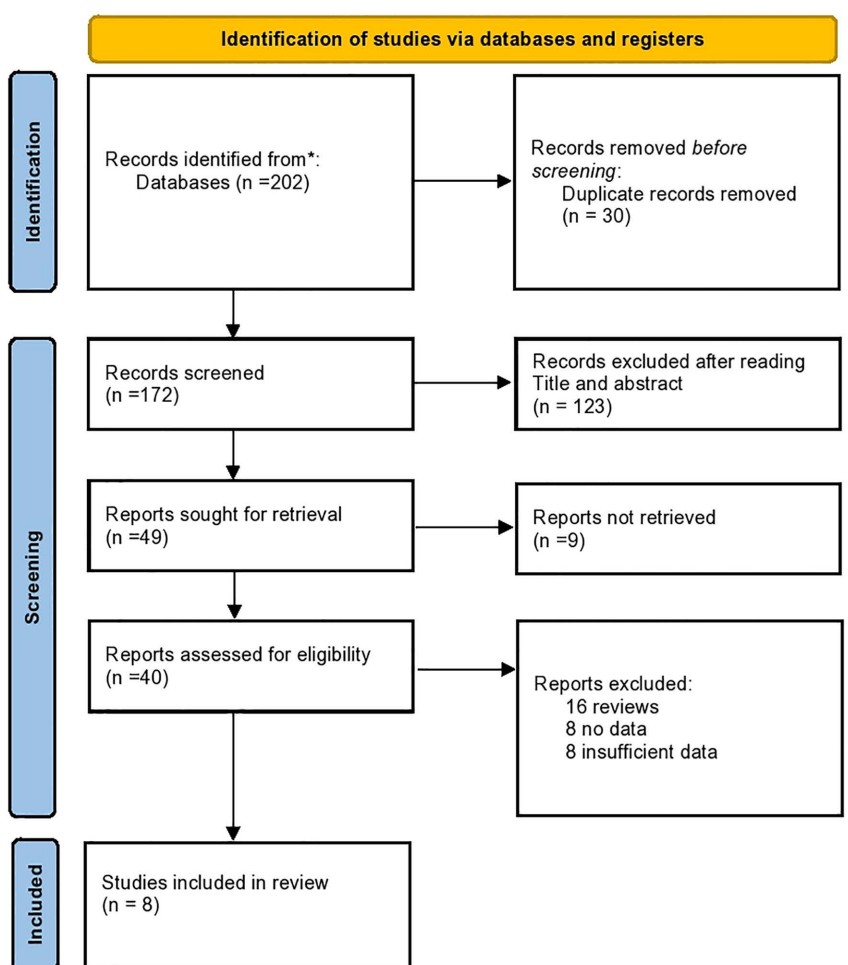

*Consider, if feasible to do so, reporting the number of records identified from each database or register searched (rather than the total number across all databases/registers).

**If automation tools were used, indicate how many records were excluded by a human and how many were excluded by automation tools.

*From:* Page MJ, McKenzie JE, Bossuyt PM, Boutron I, Hoffmann TC, Mulrow CD, et al. The PRISMA 2020 statement: an updated guideline for reporting systematic reviews. BMJ 2021;372:n71. doi: 10.1136/bmj.n71

For more information, visit: http://www.prisma-statement.org/

**Fig 1. Preferred Reporting Items for Systematic Reviews and Meta-Analyses (PRISMA) diagram detailing the study identification and selection process.**

multiple sexual partners of participants (including multiple marriages or sexual partners), marital or sexual history of the husbands of HPV-infected participants (including multiple marriages or sexual partners), presence of sexually transmitted infections (STIs) among HPV cases, contraceptive use among HPV-infected participants, and migration status of the husbands of HPV-positive participants (categorized as migrating within Nepal, migrating outside Nepal, or never migrating).

## Quality assessment of the studies

The quality assessments of each study identified were conducted by two authors (PP and AS) via an 11-item instrument recommended by the Agency for Healthcare Research and Quality (AHRQ) for cross-sectional studies, with eight stars or more considered high quality. Any disagreements were resolved by discussion and common consensus. The detailed quality assessment of the articles is shown in S1 Table.

## Statistical analyses

The data collected from the Google Sheets were exported, and analysis was performed via Jamovi 2.3.28. Prevalence estimates of HPV infection were calculated by pooling the study-specific estimates with 95% confidence intervals (CI) via the random effects model via Freeman–Turkey transformation of the inverse hyperbolic sine function. Heterogeneity was evaluated both visually via forest plots and via the $\chi2$ test on Cochrane's Q statistic and then quantified by calculating the I2. The heterogeneity test was considered statistically significant when $p \leq 0.05$. In this case, the data were analysed via a random effects model. In contrast, if $p > 0.05$, a fixed effects model was used to analyse the data. Subgroup analyses with respect to the smoking status of the HPV-infected participants, their educational background, the presence of multiple marriages or sexual partners among the HPV-infected participants and their husbands, cases with sexually transmitted infections (STIs) were conducted, use of contraceptives and the migration status of their husbands. By exploring these factors, we aimed to identify patterns and potential risk contributors to HPV infection.

# Results

## Study selection

The searches of PubMed, Embase and Google Scholar yielded a total of 202 studies. After being adjusted for duplicates, 172 studies remained. Of these, 123 studies were discarded because, after the abstracts were reviewed, these papers clearly did not meet the criteria. Additionally, 9 studies were discarded because the full text of the study was not available or because the paper could not be feasibly translated into English. The full texts of the remaining 40 citations were examined in more detail. It appeared that 32 studies did not meet the inclusion criteria as described. Eight studies met the inclusion criteria and were included in the review.

## Study characteristics

All eight studies ultimately selected for the review were cross-sectional studies published in English. The included studies involved 6082 participants. In all included studies, the primary outcome assessed was the prevalence of HPV infection among participants. The additional outcomes included the odds of HPV infection related to seven specific risk factors: smoking status, educational status, multiple sexual partners, a history of STIs, contraceptive use, and the migration status of the participants' husbands. The timing of the outcome assessment varied across studies. A summary of the included studies is shown in Table 1.

## Results of individual studies

Among a total of 6082 participants, 608 were identified as having HPV infection. The heterogeneity test revealed that the eligible studies were heterogeneous (I2 = 91.86%; p < 0.001). Hence, a random effects model was used to assess the combined incidence of HPV infection, which was 8.31% (95% CI: 5.8–10.8%), as represented in the forest plot in Fig 2.

**Table 1. Characteristics of the studies included in this systematic review and meta-analysis.**

| Author/Year | Mean age | Age at first marriage | Sampe Size | Number of cases |
|---|---|---|---|---|
| Thapa et al. 2018 [6] | 32.6±8.6 | 16.7±3.8 | 998 | 115 |
| Derek et al. 2014 [7] | 33.8±8.8 | 17.3±2.6 | 261 | 25 |
| Sherpa et al. 2010 [8] | 34.68±12.05 | 17±2.96 | 932 | 57 |
| Shakya et al. 2018 [9] | 40 | 18 | 1498 | 214 |
| Shakya et al. 2016 [2] | 40 | 18 | 1289 | 102 |
| Bhatta et al. 2017 [10] | 39.5±8.1 | 19.2±4.0 | 542 | 46 |
| Shrestha et al. 2023 [11] | 31.17±5.57 | N/A | 199 | 6 |
| Johnson et al. 2015 [12] | 33.9±8.8 | N/A | 363 | 20 |

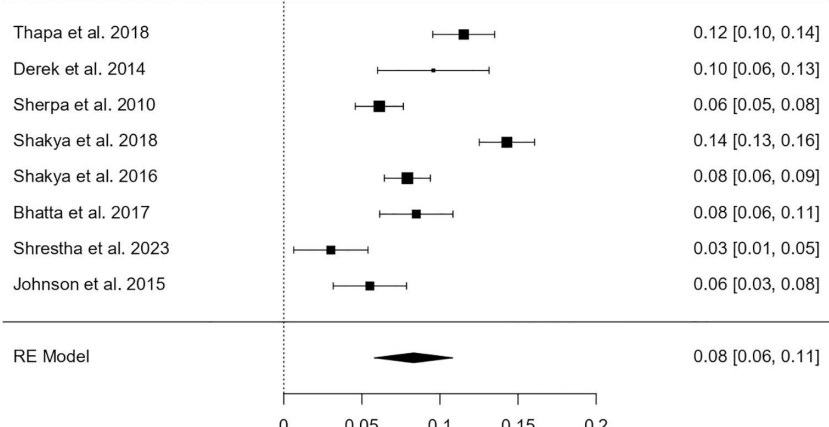

**Fig 2. Forest plot showing the estimated pooled prevalence of HPV infection in Nepal.**

The combined prevalence of HPV genotype 16 infection was 2.62% (95% CI: 0.006–0.046), as represented in the forest plot in Fig 3. The combined prevalence of HPV genotype 18 infection was 1.25% (95% CI: 0.005–0.020), as represented in the forest plot in Fig 4. The analysis revealed significant heterogeneity among studies ($I^2$ = 94.26%, p < .001 for genotype 16 and $I^2$ = 77.36%, p = .003 for genotype 18).

## Results of sensitivity analysis

Details of the sensitivity analysis are given in S6_Appendix. These analyses revealed that none of the individual studies influenced the pooled incidence, which ranged from 7.8% (95% CI = 5.7–9.8%) to 9.44% (95% CI = 7.3–11.6%). There was no significant change in the degree of heterogeneity, which was between 84.48% and 92.65%. The sensitivity analyses indicated that the results of the meta-analysis were reliable and stable.

## Results of syntheses

Our meta-analysis aims to synthesize available evidence on various factors associated with HPV infection such as smoking, educational status, sexual behaviour, contraceptive use, and migration history which have been frequently explored in primary studies. Table 2 summarizes the key findings from the meta-analysis of seven such risk factors.

Seven risk factors for HPV infection were assessed through meta-analysis. Smoking showed no significant association (log OR = −0.04; 95% CI: −0.38 to 0.30), with moderate but non-significant heterogeneity, favouring a common effects

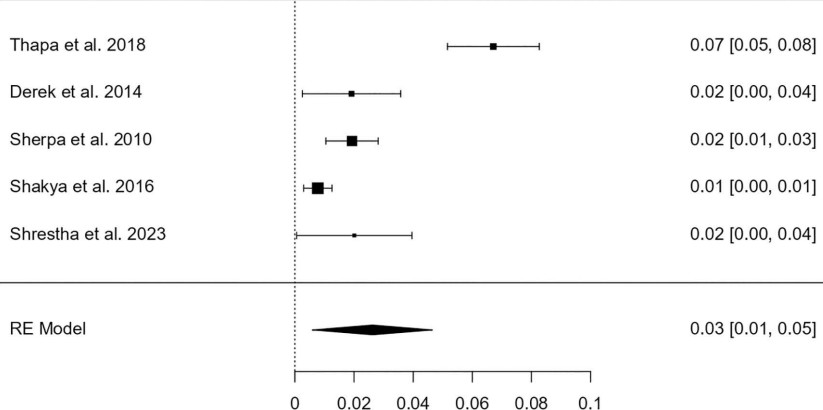

**Fig 3. Forest plot showing the estimated pooled prevalence of HPV genotype 16 infection in Nepal.**

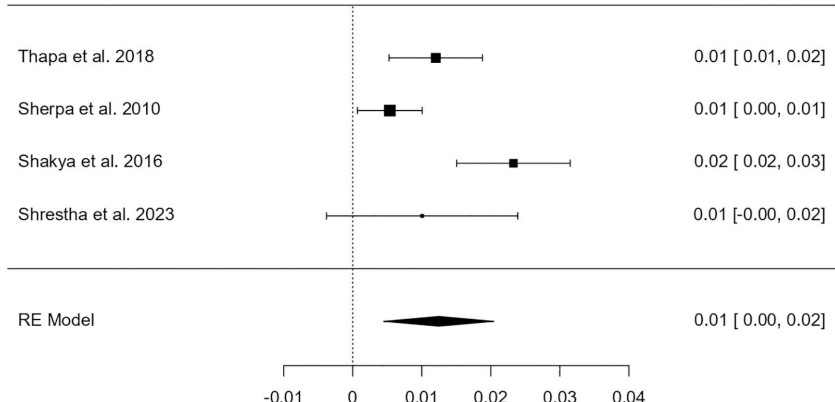

**Fig 4. Forest plot showing the estimated pooled prevalence of HPV genotype 18 infection in Nepal.**

**Table 2. Summary of pooled logarithmic odds ratios and heterogeneity estimates for risk factors associated with HPV infection.**

| Risk Factor | log(OR) | 95% CI | p-value | I² (%) | p-value (Heterogeneity) |
|---|---|---|---|---|---|
| Smoking | −0.04 | −0.38 to 0.30 | 0.65 | 49.5 | 0.14 |
| Educational Status (Literate vs Illit.) | −0.01 | −0.68 to 0.65 | 0.95 | 65.3 | 0.03 |
| Multiple Sexual Partners (Self) | 1.03 | 0.91 to 1.68 | 0.33 | N/A | N/A |
| Husband's Multiple Sexual Partners | 0.6 | 0.32 to 0.88 | 0.02 | 0 | 0.52 |
| History of STIs | 0.35 | −0.15 to 0.85 | 0.169 | 0 | <0.001 |
| Contraceptive Use | −0.32 | −1.00 to 0.33 | 0.686 | 68.1 | 0.08 |
| Migration Status | 0.26 | −0.33 to 0.85 | 0.24 | 0 | 0.74 |

model. Educational status also showed no association (log OR = −0.01; 95% CI: −0.68 to 0.65) and had significant heterogeneity ($I^2 = 65.3\%$), warranting a random effects model. Individuals with multiple sexual partners had higher odds of HPV infection (log OR = 1.03), though the association was not statistically significant (p = 0.33). In contrast, husbands' multiple sexual partners were significantly associated with HPV infection in their wives (log OR = 0.60; 95% CI: 0.32 to 0.88;

p = 0.02), with no observed heterogeneity (I² = 0%). No significant associations were found for history of STIs, contraceptive use, or migration status.

A summary of the logarithms of the odds ratios of all seven risk factors studied in this review is shown in Fig 5. Forest plot showing the estimated pooled logarithm of the odds ratio of risk factors studied for HPV infection is in S7 Appendix.

While a significant association was found for a husband's sexual behaviour, other examined risk factors did not show statistically significant relationships with HPV infection. This lack of significance does not necessarily imply absence of association, but may reflect limitations in sample size, study quality, measurement inconsistencies, and the inability of cross-sectional data to assess causality.

## Discussion

The meta-analysis pools data concerning the prevalence, genotype distribution, and risk factors associated with HPV infection among Nepalese individuals, as reported in eight cross-sectional studies that were conducted among 6,082 participants. To the best of our knowledge, this is the first meta-analysis that systematically synthesizes evidence on HPV infection in the Nepali population and thus provides useful insights into the scope of this public health challenge.

Among the 6082 participants included in this meta-analysis, 608 were identified with HPV infection, resulting in a pooled prevalence of 8.31% (95% CI: 5.8–10.8%) using a random-effects model due to substantial heterogeneity (I² = 91.86%, p < 0.001). This prevalence aligns with some regional studies but differs in magnitude when compared to global averages. For example, Bruni et al. (2010) [13], in a large-scale global meta-analysis using data from the ICO HPV Information Centre, reported an overall HPV prevalence of 11.7% among women with normal cytology, which is slightly higher than the estimate in our study. In contrast, Clifford et al. (2005) [14] reported even wider variability, with prevalence estimates ranging from 2% in Western Asia to over 20% in sub-Saharan Africa, reflecting the influence of geography, screening uptake, and sexual behavior patterns. Within South Asia, studies have generally reported HPV prevalence between 7–12%, as seen in primary studies from India [15] which support the comparability of our findings with regional data. However, some community-based studies in rural Indian populations have reported lower rates (~4–6%), likely due to conservative sexual behavior and under-detection due to limited access to sensitive testing methods.

For genotype-specific analysis, like the global scenario, HPV-16 and HPV-18 remain the most common high-risk genotypes in Nepal as well. However, the pooled prevalence of HPV-16 was 2.62% (95% CI: 0.6–4.6%), and HPV-18 was 1.25% (95% CI: 0.5–2.0%). These are slightly lower than global estimates from de Sanjosé et al. (2010) [16], who found HPV-16 and 18 in 3.2% and 1.4%, respectively, among women without cervical lesions, and much lower compared to cancer populations where HPV-16 alone is detected in 53.5% of cervical cancer cases globally. Primary studies among healthy Indian women [17] have also reported similar genotype-specific prevalence with HPV-16 ranging between 2–4%

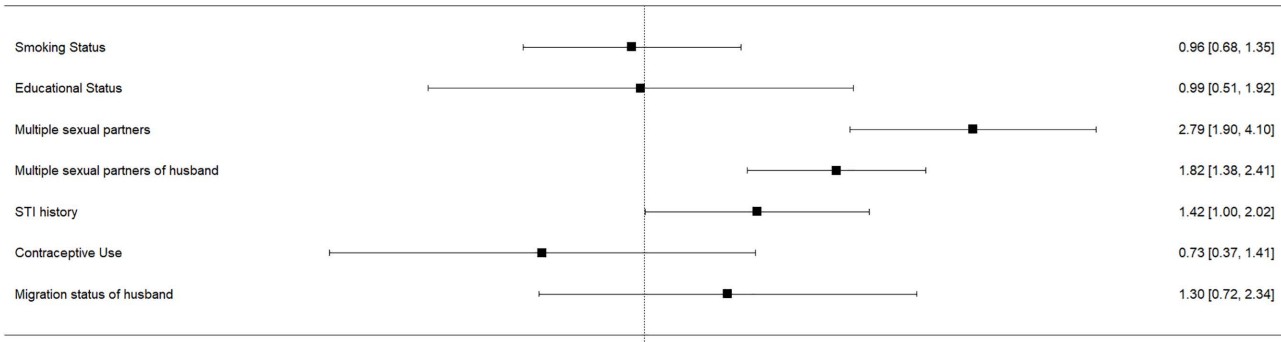

**Fig 5. Forest plot summarizing the estimated pooled logarithm of the odds ratio of the risk factors for HPV infection.**

and HPV-18 between 1–2%, which corroborates our findings. In contrast, studies from high-income countries such as the United States, as reported by Dunne et al. (2007) [18], have shown higher genotype prevalence, particularly in younger women, due to broader screening practices and more active sexual behavior in younger cohorts.

The high heterogeneity observed in genotype-specific prevalence (I² = 94.26% for HPV-16 and I² = 77.36% for HPV-18) across studies included in this meta-analysis is consistent with previous systematic reviews, such as those by Clifford et al. (2005) [14] and Bruni et al. (2010) [13], which also highlighted wide inter-study variability due to differences in age structure, population setting (urban vs rural), laboratory assays, and sexual health practices. Some studies included only married women or had limited age ranges, while others varied in whether clinician-collected or self-collected samples were used, further contributing to the heterogeneity. While the direction of estimates is similar across studies, the variation in magnitude emphasizes the need for context-specific public health strategies.

These findings reinforce the global understanding that HPV-16 and 18 are the most prevalent high-risk genotypes and are appropriately targeted by current vaccines. The pooled prevalence estimates provide baseline data for evaluating HPV vaccination impact in South Asia. As several countries, including Nepal and India, have recently incorporated HPV vaccination into their national immunization schedules [19], continuous surveillance and comparison with pre-vaccine baseline prevalence, such as the one established in this analysis, will be critical for monitoring program effectiveness.

Several findings from this review are in agreement with existing literature. The analysis revealed a significant association between husbands having multiple sexual partners and an increased risk of HPV infection in their wives. This supports global findings that highlight the importance of spousal sexual behavior in HPV transmission. Although higher odds were observed among those with multiple sexual partners, the strength of this association varied due to limitations in statistical precision. Other investigated risk factors—including smoking, educational status, history of STIs, and contraceptive use—were not found to be significantly associated with HPV infection in this review. Moderate variability across studies, particularly regarding educational status, suggests these findings should be interpreted with caution. Regarding contraceptive use, our analysis did not find a significant association with HPV infection, which is consistent with Coker et al. (2001) [20], who found no link between hormonal contraceptive use and cervical lesions after adjusting for HPV status. While our study did not find smoking to be a significant risk factor, Bowden et al. (2023) [21] noted suggestive evidence for an association with HPV incidence, though the strength of the evidence was limited by study quality. On educational status, our findings showed no significant association with HPV infection, which contrasts with Thanasas et al. (2022) [22], who demonstrated that higher socioeconomic and educational indicators were associated with greater HPV awareness and vaccine acceptance. These comparisons suggest that while some behavioral risk factors, such as partner sexual behavior, are consistently linked to HPV infection, others—like smoking, contraception, and education—may influence awareness and preventive practices more than infection rates themselves, and may vary across populations and study designs.

The analysis revealed considerable heterogeneity across studies reflecting differences in study designs and populations; therefore, large-scale and standardized studies are needed to provide more precise estimates of the prevalence and a better understanding of the epidemiology of HPV.

The overall prevalence of HPV infection in Nepal indicates a moderate but meaningful public health burden, with HPV genotypes 16 and 18 being the most commonly identified high-risk types.

This meta-analysis has several limitations at both the study and review levels. A major limitation at the study level was that the cross-sectional nature of all included studies inhibits the inference of a causal relationship between the analysed risk factors and HPV infection. Considerable heterogeneity was observed across studies, particularly for prevalence estimates and certain risk factors such as educational status. This variability likely reflects differences in study design, population characteristics, and methodological approaches, which may limit the generalizability of the findings. Additionally, factors like migration status and contraceptive use showed high variability, potentially due to inconsistent measurement or underreporting in primary studies. Owing to differences in sampling, design, measurement, and statistical analysis, this is

not uncommon in reviews of observational studies [23]. At the level of review, studies with limited access may have been missed. In addition, the included studies may have suffered from reporting bias, which may have led to the overestimation of the findings.

While these are limitations, the strength lies in attempt to synthesize comprehensive data on the prevalence of HPV, genotype distribution, and associated risk factors in Nepal, addressing a significant gap in the regional literature. By contextualizing the findings from a local and a global perspective, this meta-analysis provides take-home messages for clinicians, policy makers, and researchers.

The findings of this study have critical implications for stakeholders. These results remind healthcare workers that the emphasis needs to be on prevention, which involves not only appropriate vaccination but also education regarding behavioural risks such as spousal sexual history. Policymakers can use such insights to plan resource allocation with regard to HPV vaccination and screening programs in rural and underserved areas. Importantly, the public should be sensitive to safe sexual practices, regular screening, and prevention strategies with respect to HPV infection. This meta-analysis provides a very firm basis for future research and public health interventions for combating HPV in Nepal.

## Supporting information

**S1 Appendix.  Detailed search strategy used in the current systematic review and meta-analysis.**
(DOCX)

**S2 Appendix.  Data Set.**
(DOCX)

**S3 Appendix.  Quality assessment of the included articles via the Agency for Healthcare Research and Quality (AHRQ) Checklist.**
(DOCX)

**S4 Appendix.  PRISMA 2020 item checklist.**
(PDF)

**S5 Appendix.  PRISMA 2020 Abstracts checklist.**
(PDF)

**S6 Appendix.  Result of sensitivity analyses.**
(DOCX)

**S7 Appendix.  Forest plot showing the estimated pooled logarithm of the odds ratio of the risk factors studied for HPV infection a.** Smoking b. Educational status c. Multiple sexual partners (self) d. Multiple sexual partners of husbands e. History of sexually transmitted infections (STIs) f. Contraceptive use g. Migration status.
(PDF)

## Author contributions

**Conceptualization:** Prayash Paudel.

**Data curation:** Asutosh Sah.

**Formal analysis:** Prayash Paudel, Asutosh Sah.

**Funding acquisition:** Prayash Paudel.

**Investigation:** Prayash Paudel, Asutosh Sah.

**Methodology:** Prayash Paudel, Asutosh Sah.

**Project administration:** Prayash Paudel.

**Resources:** Prayash Paudel.

**Software:** Prayash Paudel.

**Supervision:** Prayash Paudel.

**Validation:** Prayash Paudel, Asutosh Sah.

**Visualization:** Prayash Paudel, Asutosh Sah.

**Writing – original draft:** Prayash Paudel, Asutosh Sah.

**Writing – review & editing:** Prayash Paudel, Asutosh Sah.

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
