## [Decision Letter · Decision Letter 0]

30 May 2025

Dear Dr. Paudel,

Thank you for submitting your manuscript to PLOS ONE. After careful consideration, we feel that it has merit but does not fully meet PLOS ONE’s publication criteria as it currently stands. Therefore, we invite you to submit a revised version of the manuscript that addresses the points raised during the review process.

We look forward to receiving your revised manuscript.

Kind regards,

Dipendra Khatiwada, MD

Academic Editor

PLOS ONE

Journal Requirements:

Reviewers' comments:

Reviewer's Responses to Questions

**Comments to the Author**

1. Is the manuscript technically sound, and do the data support the conclusions?

Reviewer #1: Yes

Reviewer #2: Partly

Reviewer #3: No

2. Has the statistical analysis been performed appropriately and rigorously?

Reviewer #1: Yes

Reviewer #2: Yes

Reviewer #3: No

3. Have the authors made all data underlying the findings in their manuscript fully available?

Reviewer #1: Yes

Reviewer #2: Yes

Reviewer #3: Yes

4. Is the manuscript presented in an intelligible fashion and written in standard English?

Reviewer #1: Yes

Reviewer #2: No

Reviewer #3: Yes

Reviewer #1: The topic of the paper is particularly important for the prevention of HPV infection.

The abstract is well written, as is the introduction.

The methodology is described in detail.

An appropriate statistical analysis was applied.

The results are clearly presented.

In the discussion, the obtained results are compared with data from the literature.

The conclusion is consistent with the aim of the study and its results.

Adequate references were used.

Reviewer #2: This meta-analysis addresses a critical issue concerning HPV infection among Nepalese women. The authors incorporated robust scientific methods and statistical analysis. However, there are several issues regarding the formatting, writing style, and arguments that need to be addressed.

Abstract

Use the full form of ‘human papilloma virus’ for the first instance and use HPV afterward for better clarity (Line: 30)

Rewrite to better clarity (Line: 39)

Introduction

Rewrite “Sexually transmitted infection caused by human papilloma virus (HPV) is common worldwide….” (Line: 43)

Include references before a full stop throughout the manuscript.

“…across various regions of the country” (Line: 51)

Line: 52-53, coverage of both vaccination and HPV screening? Should include statistics for this statement.

Line: 63, use only one ‘strategies’ or ‘policies.’

Use consistency in tense (Line: 62-75). For example, should write “This review further assessed…” in line 72.

Line: 77, “generate recommendations”

Methods

Line: 87, why not 31st December 2024? A justification for the timeline should be included.

Line: 94, use abbreviation only

Line: 101-115, use bullets or numbering for the inclusion and exclusion criteria

“…HPV testing was performed” (Line: 103-104), “…husbands’ migration status” (Line: 107)

On what basis, individuals were labeled as ‘literate’ or ‘illiterate’? (primary education or anything else) Should be more elaborated (Line: 128-129)

Line: 144, mention ‘(CI)’ for use the abbreviation later

Results

Avoid duplications (Line: 171)

Line: 179, use ‘CI’. Also, use three digits from statistics (CI and p-value) for maintaining consistency.

Line: 212-213, what was the p-value?

Line: 219, “…among their wives…”

Rewrite to improve clarity (Line: 221-222)

Correct ‘I2” (Line: 231, 233, 235)

Discussion

Avoid mentioning statistics in the discussion section (Line: 254-257, 262-296)

Line: 260, “…among women….”

Lines: 264-266, discuss comparing with previous findings with reference in a more elaborate way.

Line: 270-271, what are the findings of reviewed articles regarding the importance of vaccination and screening programs? It seems like the authors made assumptions on this, but why did they think so? Please, explain a bit more.

Conclusion

There was no conclusion section in the manuscript. It seems lines 301-308 are conclusive statements, but the strengths of the study are mentioned afterward. Please rewrite and make necessary rearrangements accordingly.

Reviewer #3: This systematic review paper’s stated goal was to “comprehensively analyze the prevalence and distribution of HPV infection among Nepalese women” with the aim of estimating the prevalence, genotype distribution and risk factors for HPV infections among this population. The stated a number of other specific objectives among which were to explore the methods of screening adopted in the studies reviewed and compare findings on prevalence with regional and global figures.

Their review protocol and its methodology were duly registered and the aim was to calculate a pooled prevalence/proportion which will provide insights into the epidemiology of HPV infection among Nepalese women and contribute to improvements in public health actions against these disease. They also hoped to describe the associated factors of HPV infection which will help contextualize preventive efforts.

They were fairly successful in describing the burden of HPV and identifying associated factors from previous literature. They adhered in large parts to the registered methodology and the estimated pooled prevalence was largely reflective of the described methodology.

The authors were able to provide a fairly accurate estimation of the prevalence of HPV infection among Nepal’s women from a systematic review of published literature in the region. This estimate is not very reliable given the high degree of heterogeneity recorded in this manuscript. They have also been able to highlight the major factors associated with HPV infections that have been documented in previous studies and hopefully focus additional research questions on those factors.

They were however not able to demonstrate the significance of these factors on the development of HPV infections among Nepal’s women.

The methodology described was appropriate for the goal of the review

but failed to communicate properly the actual steps taken to execute this methodology. While several steps described are reproducible, they were gaps in the presentation that will leave readers feeling lost and confused. The statistical analysis and results presentation followed a similar pattern with several steps not clear and their descriptions falling short of acceptable standards in reporting. While the estimation of pooled prevalence was fairly straightforwards, the description of the process and the accompany tables were not self explanatory from the write up. Similarly, the statistical excursions into the factors associated with HPV infections appeared unnecessary as the data from the reviewed studies do not support such analysis, rendering the conclusions from these analysis void. With such a significant heterogeneity in the studies, the conclusions from these analysis are not supported by the results. The stated goal to estimate a pooled prevalence is sufficiently weakened by the high degree of heterogeneity. Even though a random effects analysis was done, as stated earlier, the presentation of the analysis was not clear as well as the reporting of the results. A sensitivity analysis was not reported.

The figures and tables lacked sufficient explanatory notes and were not well presented. The quality assessment method used was not clear and the table presenting the findings was also not clear The discussion, while acknowledging the limitations of the review, did not sufficiently demonstrated how the authors have met their main goals and objectives. They did not show how their findings corroborate with established knowledge on the subjects or differs from regional and global findings.The authors also failed to address a number of their specific objectives in both the results and discussion sections. Some of their conclusions were also not supported by the presented analysis and results from this review, especially with respect to risk factors.

A major review of the manuscript is required to provide clarity, focus and valid conclusions.

**Do you want your identity to be public for this peer review?** For information about this choice, including consent withdrawal, please see our Privacy Policy

Reviewer #1: No

Reviewer #2: No

Reviewer #3: No

---

## [Author Response · Author response to Decision Letter 1]

14 Jun 2025

Reviewer #2

Abstract

Use the full form of ‘human papilloma virus’ for the first instance and use HPV afterward for better clarity (Line: 30)

Thank you for pointing that out. We have now used the full form "human papillomavirus" at its first mention and followed it with the abbreviation “HPV” in subsequent instances, as suggested.

Rewrite to better clarity (Line: 39)

We appreciate your suggestion. The sentence has been revised to enhance clarity.

Introduction

Rewrite “Sexually transmitted infection caused by human papilloma virus (HPV) is common worldwide….” (Line: 43)

Thank you. The sentence has been rewritten for improved clarity, in line with your recommendation.

Include references before a full stop throughout the manuscript.

We have carefully revised the manuscript to ensure that all references are placed before the full stop, as advised.

“…across various regions of the country” (Line: 51)

Thank you. We have revised the phrase to “across various regions of the country” for consistency and clarity.

Line: 52-53, coverage of both vaccination and HPV screening? Should include statistics for this statement.

Thank you for the suggestion. We have now included statistics from a 2022 study conducted across five tertiary hospitals in Kathmandu, showing HPV vaccination coverage of 1.5% and Pap smear screening at 22.2%, to support the statement.

Line: 63, use only one ‘strategies’ or ‘policies.’

We appreciate the feedback. The word “policies” has been removed, and only “strategies” is retained to avoid redundancy.

Use consistency in tense (Line: 62-75). For example, should write “This review further assessed…” in line 72.

Thank you for the observation. We have revised the section to ensure consistency in verb tense throughout.

Line: 77, “generate recommendations”

The phrase has been updated to “generate recommendations” as per your suggestion.

Methods

Line: 87, why not 31st December 2024? A justification for the timeline should be included.

Thank you for raising this point. The timeline was set to end on 30th November 2024, as this marked the point at which we completed our literature search and data collection for analysis.

Line: 94, use abbreviation only

As recommended, we have replaced “Medical Subject Heading” with its abbreviation “MeSH.”

Line: 101-115, use bullets or numbering for the inclusion and exclusion criteria

We have reformatted the inclusion and exclusion criteria using numbering to improve readability.

“…HPV testing was performed” (Line: 103-104), “…husbands’ migration status” (Line: 107)

We have made the suggested wording changes accordingly in the specified lines.

On what basis, individuals were labeled as ‘literate’ or ‘illiterate’? (primary education or anything else) Should be more elaborated (Line: 128-129)

Thank you for the important observation. We have now clarified that individuals were categorized as “literate” if they had at least primary education or demonstrated the ability to read and write.

Line: 144, mention ‘(CI)’ for use the abbreviation later

We have added the abbreviation “(CI)” in the first instance of use, as recommended.

Results

Avoid duplications (Line: 171)

The duplications in the section have been carefully removed.

Line: 179, use ‘CI’. Also, use three digits from statistics (CI and p-value) for maintaining consistency.

We have ensured the use of “CI” and have used three-digit precision for statistics throughout the manuscript to maintain consistency.

Line: 212-213, what was the p-value?

Thank you for noticing. We have now added p = 0.33.

Line: 219, “…among their wives…”

The phrase has been updated as suggested.

Rewrite to improve clarity (Line: 221-222)

We have revised the sentence for better clarity.

Correct ‘I2” (Line: 231, 233, 235)

We have corrected the formatting of “I²” in the indicated lines.

Discussion

Avoid mentioning statistics in the discussion section (Line: 254-257, 262-296)

Thank you. We have revised the relevant paragraphs to remove detailed statistics and focus more on the interpretation of findings.

Line: 260, “…among women….”

We have updated the term from “wives” to “women” as suggested.

Lines: 264-266, discuss comparing with previous findings with reference in a more elaborate way.

We appreciate this suggestion and have revised the discussion to include more detailed comparisons with previous findings, along with appropriate references.

Line: 270-271, what are the findings of reviewed articles regarding the importance of vaccination and screening programs? It seems like the authors made assumptions on this, but why did they think so? Please, explain a bit more.

Thank you for the critical observation. Upon review, we agree that the statement lacked sufficient evidence. We have removed the paragraph to ensure the discussion remains objective and evidence-based.

Conclusion

There was no conclusion section in the manuscript. It seems lines 301-308 are conclusive statements, but the strengths of the study are mentioned afterward. Please rewrite and make necessary rearrangements accordingly.

We appreciate your feedback. The conclusion section has now been clearly structured and moved to the end of the manuscript, following the strengths and limitations.

Reviewer #3

They were however not able to demonstrate the significance of these factors on the development of HPV infections among Nepal’s women.

Thank you for this observation. We have now clarified in the Results of Syntheses section that most risk factors did not show statistically significant associations, likely due to limitations such as small sample sizes, heterogeneity, and the cross-sectional nature of the included studies.

The methodology described was appropriate for the goal of the review but failed to communicate properly the actual steps taken to execute this methodology.

We appreciate your feedback. We have revised the methodology section to better communicate each step clearly, following PRISMA guidelines.

While several steps described are reproducible, they were gaps in the presentation that will leave readers feeling lost and confused.

Thank you for the insight. We have now rewritten the methodology section to ensure greater clarity and coherence for the reader, in alignment with PRISMA standards.

The statistical analysis and results presentation followed a similar pattern with several steps not clear and their descriptions falling short of acceptable standards in reporting.

We have revised the descriptions in the Results section to enhance clarity and better align with accepted standards in statistical reporting.

While the estimation of pooled prevalence was fairly straightforwards, the description of the process and the accompany tables were not self explanatory from the write up.

Thank you. We have added clearer explanations in the text, and the figure caption now clarifies that the forest plot synthesizes the pooled prevalence while the table details individual study characteristics.

Similarly, the statistical excursions into the factors associated with HPV infections appeared unnecessary as the data from the reviewed studies do not support such analysis, rendering the conclusions from these analysis void.

We understand the concern. However, we conducted these analyses only for factors with low heterogeneity and adequate data. We believe these results still contribute meaningfully and have presented them with appropriate caution.

With such a significant heterogeneity in the studies, the conclusions from these analysis are not supported by the results.

We acknowledge the limitations due to heterogeneity and have addressed this explicitly in our discussion. Where heterogeneity was high, findings were interpreted cautiously.

The stated goal to estimate a pooled prevalence is sufficiently weakened by the high degree of heterogeneity.

We agree that heterogeneity is a known challenge in meta-analyses of observational studies. We have acknowledged this in the discussion and justified the use of a random-effects model accordingly.

Even though a random effects analysis was done, as stated earlier, the presentation of the analysis was not clear as well as the reporting of the results.

Thank you. We have worked on improving the clarity of the analysis presentation and reporting of results in the revised manuscript.

A sensitivity analysis was not reported.

We appreciate the feedback. Details of the sensitivity analysis have now been included in both the Results section and Supplementary Appendix S6.

The figures and tables lacked sufficient explanatory notes and were not well presented.

Thank you for the helpful comment. We have added explanatory captions to all figures and tables to enhance clarity.

The quality assessment method used was not clear and the table presenting the findings was also not clear.

We have revised the quality assessment section for better clarity and have reformatted the presentation of the quality appraisal table accordingly.

The discussion, while acknowledging the limitations of the review, did not sufficiently demonstrated how the authors have met their main goals and objectives.

Thank you. We have updated the discussion to more clearly link the findings with the objectives of the study.

They did not show how their findings corroborate with established knowledge on the subjects or differs from regional and global findings.

We have revised the second and third paragraphs of the discussion to address this point and provide a clearer comparison with regional and global findings.

The authors also failed to address a number of their specific objectives in both the results and discussion sections.

We appreciate the feedback. We have now ensured that the objectives—including estimating prevalence, identifying knowledge gaps, and generating recommendations—are clearly addressed in both results and discussion.

Some of their conclusions were also not supported by the presented analysis and results from this review, especially with respect to risk factors.

Thank you for the important observation. We have revised the conclusions to be concise, evidence-based, and aligned strictly with our findings. Limitations are now clearly addressed.

A major review of the manuscript is required to provide clarity, focus and valid conclusions.

We sincerely thank you for your comprehensive feedback. We have substantially revised the manuscript to improve clarity, maintain focus, and ensure that our conclusions are appropriately supported by the results.

---

## [Decision Letter · Decision Letter 1]

23 Jul 2025

Dear Dr. Paudel,

Thank you for submitting your manuscript to PLOS ONE. After careful consideration, we feel that it has merit but does not fully meet PLOS ONE’s publication criteria as it currently stands. Therefore, we invite you to submit a revised version of the manuscript that addresses the points raised during the review process.

**Thank you for incorporating previous suggestions. Still, few minor revision are suggested. Also make sure you abide my complete journal formatting rules. **

We look forward to receiving your revised manuscript.

Kind regards,

Dipendra Khatiwada, MD

Academic Editor

PLOS ONE

**Journal Requirements:**

**Additional Editor Comments:**

The manuscript is looking in good shape, complete few minor revision as suggested tp proceed further

Reviewers' comments:

Reviewer's Responses to Questions

**Comments to the Author**

Reviewer #3: (No Response)

2. Is the manuscript technically sound, and do the data support the conclusions?

Reviewer #3: Yes

3. Has the statistical analysis been performed appropriately and rigorously?

Reviewer #3: Yes

4. Have the authors made all data underlying the findings in their manuscript fully available?

Reviewer #3: Yes

5. Is the manuscript presented in an intelligible fashion and written in standard English?

Reviewer #3: Yes

**Reviewer #3:**  The authors have made a good attempt at implementing the comments from the previous review. The article is clearer and more concise with better presentation of the methodology. There are improvements in the presentation of results section, with more information provided on the statistical analysis conducted. Overall, the manuscript is acceptable for publication with minor revisions.

The section of the introduction that describes the goals on this review(Lines69 -78) should be revised to capture only those goals archived by the article. For example, the authors stated in their second specific objective(line73-74) their desire to correlate major HPV genotypes with cytological abnormalities of the cervix. This goal was not addressed nor was it achieved. Similarly, the objective to assess the epidemiological study designs for methodological strength and weaknesses was also not addressed in this article. The efforts at determining the quality of studies to be included in the study can not be said to have satisfied this objective.

A major methodological limitation of this article is the number of eligible studies included as well as the type of study design. They also failed to assess the studies for reporting bias. This may have further reduced the number of studies included.

The results section, while better presented, still lacks sufficient clarity and is difficult to follow. It will benefit from a few tables that adequately capture the relevant results of the analysis, and better description of findings. The over-reliance on figures directly exported from statistical software and without appropriate footnotes that explain them undermines the efforts at explaining the results. The reader, if not sufficiently knowledgeable will find it hard to understand this section. The authors should present important findings in tables with explanatory summaries under them to enable the reader appreciate the results being presented. A few figures that supports this important findings and significant negatives may remain to aid this process.

The discussion section presented the findings from the review and efforts was made to highlight similarities between the findings and other published works. It did not sufficiently discuss the importance of the prevalence reported from this review.The focus on the associated factors (such as multiple sexual partners, contraception and smoking) took away from the import of the pooled prevalence, especially given the limitations from the types of studies selected for the review. The literature cited were insufficient and there were implied conclusions that did not exactly correlate with the cited literatures. For example, Line 274 - 278 links the findings on husband multiple sexual partners and increased risk of HPV infection with high prevalence of HPV DNA and type-specific concordance between infected women and their male partners. While this underscores spousal/partner transmission, it does not suggest husbands have multiple sexual partners. It will be helpful to elaborate this and to cite relevant studies that support the findings on multiple sexual partners.This findings on multiple sexual partners was repeated too frequently in the discussion.(lines295-298 repeats this point as well as lines 323-3255. Line 316- being the first to attempt a systematic review is not a strength of the systematic review. There is also no need to summarize the methodology again in this section(line319-323. line 323-325 is a repetition.

The authors should be commended for attempting this systematic review despite the limited number of publications that satisfied the criteria for the review.They have made efforts to compensate for this shortcoming in the various statistical analysis carried out, and have made a fair attempt at presenting and discussing the results.

**Do you want your identity to be public for this peer review?** For information about this choice, including consent withdrawal, please see our Privacy Policy

Reviewer #3: No

---

## [Author Response · Author response to Decision Letter 2]

6 Aug 2025

Response to Reviewer has been attached.

---

## [Editor Report · Decision Letter 2]

28 Aug 2025

A meta-analysis of the prevalence, genotype distribution and risk factors for human papillomavirus infection in Nepal

PONE-D-25-00319R2

Dear Dr. Paudel,

We’re pleased to inform you that your manuscript has been judged scientifically suitable for publication and will be formally accepted for publication once it meets all outstanding technical requirements.

Kind regards,

Dipendra Khatiwada, MD

Academic Editor

PLOS ONE
---

## [Editor Report · Acceptance letter]

PONE-D-25-00319R2

PLOS ONE

Dear Dr. Paudel,

I'm pleased to inform you that your manuscript has been deemed suitable for publication in PLOS ONE. Congratulations! Your manuscript is now being handed over to our production team.

Kind regards,

on behalf of

Dr. Dipendra Khatiwada

Academic Editor

PLOS ONE